

# Comparative microscopic study of entropies and their production

Philipp Strasberg⋆ and Joseph Schindler

Física Teòrica: Informació i Fenòmens Quàntics, Departament de Física, Universitat Autònoma de Barcelona, 08193 Bellaterra (Barcelona), Spain

⋆ philipp.strasberg@uab.cat

## Abstract

We study the time evolution of eleven microscopic entropy definitions (of Boltzmann-surface, Gibbs-volume, canonical, coarse-grained-observational, entanglement and diagonal type) and three microscopic temperature definitions (based on Boltzmann, Gibbs or canonical entropy). This is done for the archetypal nonequilibrium setup of two systems exchanging energy, modeled here with random matrix theory, based on numerical integration of the Schrödinger equation. We consider three types of pure initial states (local energy eigenstates, decorrelated and entangled microcanonical states) and three classes of systems: (A) two normal systems, (B) a normal and a negative temperature system and (C) a normal and a negative heat capacity system. We find: (1) All types of initial states give rise to the same macroscopic dynamics. (2) Entanglement and diagonal entropy sensitively depend on the microstate, in contrast to all other entropies. (3) For class B and C, Gibbs-volume entropies can violate the second law and the associated temperature becomes meaningless. (4) For class C, Boltzmann-surface entropies can violate the second law and the associated temperature becomes meaningless. (5) Canonical entropy has a tendency to remain almost constant. (6) For a Haar random initial state, entanglement or diagonal entropy behave similar or identical to coarse-grained-observational entropy.

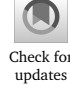

# 1  Motivation

In view of the pivotal roles of entropy, temperature and the second law in science, it is surprising that more than 150 years after their inception [1] scientists have not even approximately agreed on their basic microscopic definitions. Instead, different schools of thought have formed that differ widely both from a philosophical-conceptual and (what will be our focus here) quantitative-numerical point of view. In addition, the topic appears unnecessarily mystified as evidenced, for instance, by the often quoted statement *"no one knows what entropy really is"*—allegedly made by von Neumann while discussing with Shannon about a name for his "entropy" concept [2], even though it is unclear whether von Neumann truly said it [3]. This quote also seems particularly unfortunate as historical evidence indicates that von Neumann had a clear opinion on which microscopic entropy definition to use in statistical physics [4–6], which—to add to the confusion—is *not* what is nowadays known as von Neumann entropy.

For a long time these differences could be neglected in practice because numerical discrepancies quickly vanish for normal macroscopic systems.[1] However, various current research directions related to long range interactions [10] arising in gravitating systems [11–14] and quantum many-body systems [15], the origin of the arrow of time [16–19], isolated quantum systems with cold atoms and trapped ions [20–22], nanoscale systems [23–25], among others, make it necessary to reconsider foundational questions in statistical mechanics, and also allow to test them experimentally (see, e.g., Refs. [26–31]). Thus, it seems as if the biggest strength of statistical mechanics—namely, its "theory invariance" for normal macroscopic systems— now turns against it as schools of thought are already established.

With advances in computer technologies it has now become possible to simulate isolated quantum (and also classical) systems, which are large enough to exhibit thermodynamic behaviour, without approximations. Consequently, a number of interesting case studies emerged that focused on the nonequilibrium dynamics of thermodynamic entropy from first principles [32–50].[2] However, somewhat echoing the criticism above, a study critically comparing a variety of entropy definitions on the same footing is not known to us.

The primary goal of this study is to close this gap by providing a *comparative* study of many different entropy notions, and to motivate researchers to take this problem seriously. To do so, we start by briefly introducing the model and the various definitions in Sec. 2. Section 3 then presents our numerical results. Finally, all pertinent observations are summarized in Sec. 4.

To avoid any bias in the presentation that favors a particular entropy, our strategy is to exclusively focus on indisputable numerical results that follow from numerically exact integration of the Schrödinger equation. In particular, we do *not* mention any conceptual issues or mathematical properties related to the various entropies. While we believe they are important, they are already well covered in the literature that we cite and, apparently, theoretical and analytical arguments do not seem to have convinced the respective opponents yet. Moreover, our numerical results are transparent insofar that confirming them does not require very advanced coding techniques or supercomputers. Finally, the reader will also see that the results we present below are rather generic in the sense that we did not use any fine tuning or exhaustive search to generate them.

---

[1]We call a system *normal* if it has a concave *and* non-decreasing Boltzmann entropy as a function of energy. In the literature one often requires only concavity, which ensures a positive heat capacity and implies equivalence of ensembles [7–9], but it is more convenient here to have a separate category for systems that can show negative (Boltzmann) temperatures.

[2]The literature is even wider if one includes equilibrium systems or effective dynamics such as master equations. This is not our focus here because both cases potentially mask a large part of the problem.

## 2 Framework

**Setup**

Our idea is to study a paradigmatic nonequilibrium process, namely the flow of heat between two bodies $A$ and $B$ (similar but simpler versions of the model below have been studied, e.g., in Refs. [51–53]). Each system $X \in \{A, B\}$ is modeled with a Hilbert space $\mathcal{H}_X$ with dimension $D_X$ and Hamiltonian $H_X = \sum_{j=1}^{D_X} \epsilon_j^X |\epsilon_j\rangle\langle\epsilon_j|_X$.

Before specifying the interaction, we look at each system separately and drop for now the label $X$ for notational simplicity. The eigenenergies $\epsilon_j$ are distributed according to a density of states (DOS) $\mu(\epsilon)$ such that $\mu(\epsilon)d\epsilon$ are the number of microstates in an energy interval $[\epsilon, \epsilon + d\epsilon)$. Since $\mu(\epsilon)$ can be arbitrary, there is no assumption here, but below we consider three classes of models:

A Normal systems where $\ln\mu(\epsilon)$ is concave and monotonically increasing (we choose a square root dependence below).

B Negative temperature systems where $\ln\mu(\epsilon)$ is concave but not monotonically increasing for all $\epsilon$ (we choose a Gaussian distribution below).

C Negative heat capacity systems where $\ln\mu(\epsilon)$ is convex (we choose a quadratic dependence below).

Many entropies require a coarse graining of the energies. To this end, we first restrict the discussion to a fixed energy interval $[E_{\min}, E_{\max}]$, where $E_{\min}$ is the ground state energy and $E_{\max}$ some suitable high energy cutoff. This interval is then divided into $M$ equidistant subintervals of size $\delta_E = (E_{\max} - E_{\min})/M$. The coarse grained energies, for which we choose a capital letter, can thus be defined as $E_\alpha = E_{\min} + (\alpha - 1/2)\delta_E$ with $\alpha \in \{1, \ldots, M\}$. They label the subintervals $I_\alpha = [E_\alpha - \delta_E/2, E_\alpha + \delta_E/2)$ such that $[E_{\min}, E_{\max}] = \bigcup_\alpha I_\alpha$. The corresponding projector is denoted $\Pi_\alpha = \sum_{\epsilon_j \in I_\alpha} |\epsilon_j\rangle\langle\epsilon_j|$.

To match the fine and coarse descriptions, we demand that the number of microstates $W_\alpha = \text{tr}\{\Pi_\alpha\} \in \mathbb{N}$ compatible with the coarse energy $E_\alpha$ samples the DOS while respecting a fixed Hilbert space dimension $D$ (this requires some rounding). Within each coarse window we then distribute the eigenenergies $\epsilon_j$ evenly, but the results are quite insensitive to the question where exactly the $\epsilon_j$ lie as long as they are smeared out and the $W_\alpha$ sample well $\mu(\epsilon)$. To meet the last point, we require that $\mu(E_\alpha)/\sum_\alpha \mu(E_\alpha) \approx W_\alpha/D$ and that the Riemann sum $\sum_\alpha \delta_E W_\alpha/D$ approximates $\int \mu(\epsilon)d\epsilon / \sum_\alpha \mu(E_\alpha)$.

Finally, we restrict the dynamics to a subinterval $[E_a, E_b] \subset [E_{\min}, E_{\max}]$, which is still large enough to accommodate several coarse grained energy windows. This restriction is forced on us by numerical limitations and the fact that certain entropy and temperature definitions below require either knowledge of $\mu(\epsilon)$ down to the ground state or to match a Gibbs distribution to some mean energy. Since we want to model the exchange of energy between two systems $A$ and $B$ that are approximately equal in size, we are restricted to a local effective Hilbert space dimension of around $D_{\text{eff}} = \text{tr}\{\Pi_{[E_a, E_b]}\} \lesssim 400$, where $\Pi_{[E_a, E_b]}$ is the projector on the subinterval $[E_a, E_b]$. Thus, if we would set $[E_a, E_b] = [E_{\min}, E_{\max}]$, we would either need to restrict the discussion to a small range of coarse grained energies, i.e., close-to-equilibrium dynamics, or the local DOS would be too small to display thermodynamic behaviour. To avoid both options, we cut away parts of the Hilbert space that have little dynamical relevance. This picture is illustrated in Fig. 1 and it will become clearer when we present the definitions and numerical results.

SciPost Phys. **17**, 143 (2024)

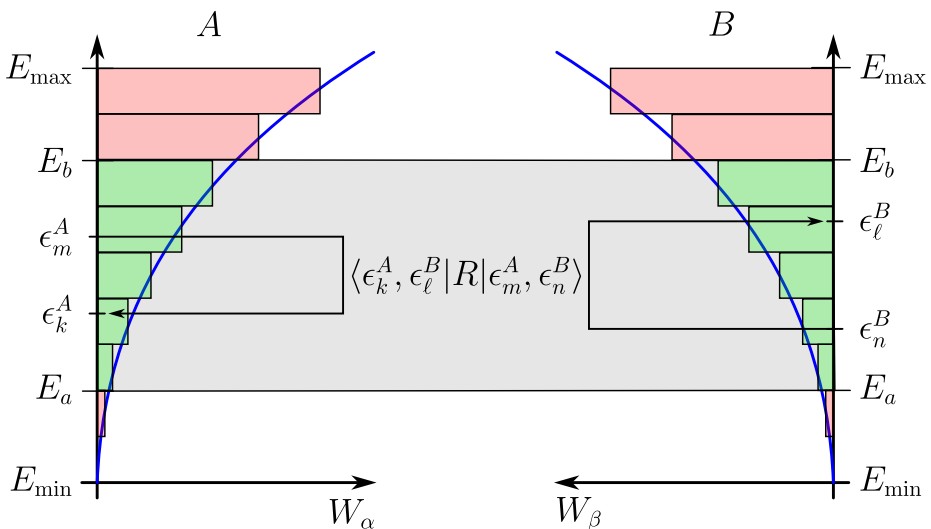

Figure 1: Sketch of two interacting systems $A$ and $B$ and their (discretized) DOS. The blue smooth line sketches the real distribution $\mu(\epsilon)$ to which we numerically fit a coarse grained distribution (symbolized by the histograms). Moreover, the dynamically relevant subinterval is indicated in green, whereas the red part is only there to properly define some particular entropies and temperatures. The dynamics is thus restricted to the grey subpart, in which we sketched an energy exchange process mediated by a random matrix interaction.

To complete the model, we specify an interaction Hamiltonian $V = \lambda R$. Here, $\lambda$ is some interaction strength (see below) and $R$ is a banded random matrix. The motivation to use random matrix theory stems from the extensive literature that has shown its power to model generic complex and non-integrable quantum systems in a minimal way [54–60]. Moreover, recalling the close connection between random matrix theory and the eigenstate thermalization hypothesis [59–64], it is reasonable to conjecture that our findings continue to hold for more realistic quantum many-body systems as well, at least qualitatively. Bandedness on $R$ is imposed by demanding $\langle \epsilon_k^A, \epsilon_\ell^B | R | \epsilon_m^A, \epsilon_n^B \rangle = 0$ if $|\epsilon_k^A + \epsilon_\ell^B - \epsilon_m^A - \epsilon_n^B| > \delta_V$, where $|\epsilon_m^A, \epsilon_n^B\rangle = |\epsilon_m^A\rangle \otimes |\epsilon_n^B\rangle$ is the local energy eigenbasis of $H_A + H_B$. All remaining elements of $R$ are uniformly filled with zero-mean-unit-variance Gaussian random numbers, except of the diagonal elements of $R$, which we set to zero.

In all numerical results below we set $E_{\min} = 0$, $E_{\max} = 2.1$, $E_a = 0.42$, $E_b = 1.4$ and $\delta_E = 0.14$. This implies $M = 15$ coarse grained energy windows in the full interval $[E_{\min}, E_{\max}]$ and seven windows in the dynamically relevant subinterval $[E_a, E_b]$. Moreover, the coupling strength $\lambda$ and the bandedness $\delta_V$ are chosen to satisfy the following demands:

1. We want a weak coupling between $A$ and $B$ such that $\langle H_A + H_B \rangle(t) \approx \langle H_A + H_B \rangle(0)$. In this regime bulk properties dominate surface properties and it becomes meaningful to talk about thermodynamic properties (energy, entropy, temperature) of $A$ or $B$ alone. Specifically, we define weak coupling by demanding that the fluctuations in the interaction energy $\langle V^2 \rangle_{\text{mic}}$, where $\langle \ldots \rangle_{\text{mic}}$ denotes a microcanonical average, are small compared to $\langle H_A + H_B \rangle_{\text{mic}}^2$. Using the bandedness of $V$, we estimate $\langle V^2 \rangle_{\text{mic}} \approx \lambda^2 D_{\text{eff}}^2 \delta_V / \Delta_E$, where $\Delta_E = E_b - E_a$ denotes the energy range of the dynamically relevant regime (not to be mixed up with the coarse graining width $\delta_E$). Moreover, we have $\langle H_A + H_B \rangle_{\text{mic}}^2 \approx 4\Delta_E^2$, tacitly assuming the arbitrary energy offset $E_a$ to be subtracted.

2. Opposite to the weak coupling requirement, we also need to demand that the energy levels of $A$ and $B$ are sufficiently strongly coupled to allow efficient energy exchanges. This implies that the coupling strength $\lambda$, which broadens the levels in an interacting system, should exceed the mean level spacing $\Delta_E/D_{\text{eff}}$ of $A$ or $B$ (this is a rough approximation assuming that all $D_{\text{eff}}$ levels are distributed evenly in $[E_a, E_b]$).

3. Finally, we need to find a compromise between a very sparse matrix with small $\delta_V$, which allows to simulate systems with larger $D_{\text{eff}}$, and a large $\delta_V$ such that the eigenstates of the global Hamiltonian $H_A + H_B + V$ are sufficiently delocalized with respect to the local energy eigenbasis of $H_A + H_B$: if $\delta_V$ is too small, energy transport becomes blocked due to Anderson localization.

What we found to work well is the choice

$$\lambda^2 = \frac{1}{D_{\text{eff}}^A D_{\text{eff}}^B}\left(\frac{1}{5}\frac{\Delta_E^2}{\delta_V}\right)^2, \quad \delta_V = \frac{\delta_E}{6}. \tag{1}$$

Finally, because a total Hilbert space dimension of $D_{\text{eff}}^A D_{\text{eff}}^B \lesssim (400)^2 = 160,000$ is too large for exact diagonalization, we use the sparsity of the Hamiltonian together with the Askar-Cakmak time propagation algorithm [65] until numerical convergence is reached. The idea of this algorithm is to transform the symmetric expression $|\psi(t + dt)\rangle - |\psi(t - dt)\rangle = (e^{-idtH} - e^{idtH})|\psi(t)\rangle$ into the propagation scheme $|\psi(t + dt)\rangle \approx -2idtH|\psi(t)\rangle + |\psi(t - dt)\rangle$. Moreover, for the first term we set $|\psi(dt)\rangle \approx (1 - idtH - dt^2 H^2/2)|\psi(0)\rangle$.

## Initial states

We consider three different types of initial nonequilibrium states. To this end, we choose some initial coarse grained energies $E_{\alpha(0)}$ and $E_{\beta(0)}$ for system $A$ and $B$ and set

1. Local energy eigenstates: $|\psi(0)\rangle = |\epsilon_m^A\rangle \otimes |\epsilon_n^B\rangle$ with randomly chosen $\epsilon_m^A \in I_{\alpha(0)}$ and $\epsilon_n^B \in I_{\beta(0)}$.

2. Decorrelated microcanonical states: $|\psi(0)\rangle \sim \Pi_{\alpha(0)}|\psi_R^A\rangle \otimes \Pi_{\beta(0)}|\psi_R^B\rangle$, where $|\psi_R^A\rangle$ is a Haar random state in $\mathcal{H}_A$ (and similarly for $B$). Note that $\Pi_{\alpha(0)}|\psi_R^A\rangle$ is a pure state version of the microcanonical ensemble.

3. Correlated microcanonical states: $|\psi(0)\rangle \sim \Pi_{\alpha(0)} \otimes \Pi_{\beta(0)}|\psi_R^{AB}\rangle$, where $|\psi_R^{AB}\rangle$ is a Haar random state in $\mathcal{H}_A \otimes \mathcal{H}_B$. Note that this state is with high probability strongly entangled.

We remark that all three states *look macroscopically the same*. By definition this mean that their coarse grained probability distributions are the same, namely

$$p_{\alpha,\beta}(0) \equiv p(E_\alpha, E_\beta; 0) = \langle\psi(0)|\Pi_\alpha \otimes \Pi_\beta|\psi(0)\rangle = \delta_{\alpha,\alpha(0)}\delta_{\beta,\beta(0)}, \tag{2}$$

with the Kronecker delta $\delta_{\alpha,\alpha'}$.

Note that we use the indices $\alpha, \alpha', \dots$ ($\beta, \beta', \dots$) to label properties of system $A$ ($B$). In favor of a more concise notation we then drop the superscripts $A$ and $B$ whenever no confusion is possible (as done here).

**Entropy and temperature definitions**

We start with the well known Boltzmann entropy $S_B(E_\alpha, E_\beta) = \ln W_\alpha + \ln W_\beta$ for a state having coarse energies $(E_\alpha, E_\beta)$. However, in general the distribution $p_{\alpha,\beta}(t)$ will not remain peaked around a single energy as in Eq. (2) and there are two options to directly generalize Boltzmann's entropy. First, we can consider the average energy $\langle E_A \rangle = \sum_\alpha E_\alpha p_\alpha(t)$ of $A$, where $p_\alpha(t) = \sum_\beta p_{\alpha,\beta}(t)$ is the marginal (and similarly for $B$), to define the *Boltzmann entropy of the average*:

$$S_B(\langle E \rangle; t) \equiv \ln W_{\langle \alpha \rangle} + \ln W_{\langle \beta \rangle}. \tag{3}$$

Here, $\langle E \rangle$ is shorthand for $(\langle E_A \rangle, \langle E_B \rangle)$ and $\langle \alpha \rangle$ is the index $\alpha$ minimizing $|E_\alpha - \langle E_A \rangle|$ (and similarly for $\langle \beta \rangle$). Alternatively, we can define the *averaged Boltzmann entropy*

$$\langle S_B \rangle(t) \equiv \sum_\alpha p_\alpha(t) \ln W_\alpha + \sum_\beta p_\beta(t) \ln W_\beta. \tag{4}$$

There is another possibility to generalize Boltzmann's entropy known as *coarse-grained* or *observational* (cgo) *entropy* by incorporating the Shannon entropy $H_{\mathrm{Sh}}(p_{\alpha,\beta})$ of the macro-distribution $p_{\alpha,\beta}$:

$$S_{\mathrm{cgo}}(t) \equiv H_{\mathrm{Sh}}[p_{\alpha,\beta}(t)] + \langle S_B \rangle(t) = \sum_{\alpha,\beta} p_{\alpha,\beta}(t) \Big[ -\ln p_{\alpha,\beta}(t) + \ln W_\alpha^A + \ln W_\beta^B \Big]. \tag{5}$$

This definition has been suggested by von Neumann [4–6], for recent introductions see Refs. [66, 67]. It is further possible to consider a *local cgo entropy* defined only in terms of local quantities of $A$ or $B$:

$$S_{\mathrm{cgo}}^\otimes(t) \equiv H_{\mathrm{Sh}}[p_\alpha(t)] + H_{\mathrm{Sh}}[p_\beta(t)] + \langle S_B \rangle(t). \tag{6}$$

Its difference $S_{\mathrm{cgo}}^\otimes - S_{\mathrm{cgo}}$ with the previous one is given by the always positive mutual information $I_{AB}(t) \equiv H_{\mathrm{Sh}}[p_\alpha(t)] + H_{\mathrm{Sh}}[p_\beta(t)] - H_{\mathrm{Sh}}[p_{\alpha,\beta}(t)] \geq 0$.

In the following, we will sometimes refer to all the Boltzmann-type entropies as "surface entropies" to distinguish them from "volume entropies", which are obtained by replacing the number of microstates $W_\alpha$ for a given energy $E_\alpha$ by

$$\Omega_\alpha = \sum_{\alpha' \leq \alpha} W_{\alpha'}. \tag{7}$$

Here, the sum runs over all $\alpha'$ such that $E_{\alpha'} \leq E_\alpha$ (and similarly we define $\Omega_\beta$). The origin of the "surface" and "volume" terminology comes from a classical phase space picture, where $W_\alpha$ is the integral over an energy shell (a surface) and $\Omega_\alpha$ is the integral over the phase space enclosed by that shell (a volume). The idea to replace $W_\alpha$ by $\Omega_\alpha$ was first considered by Gibbs [68], and it received broad attention in the recent debate about the possibility of negative temperatures [29, 69–75] without, as it seems, reaching any consensus yet. We label the entropies that result from turning the surface definitions into volume definitions by a subscript $G$ (for Gibbs):

$$S_G(\langle E \rangle; t), \quad \langle S_G \rangle(t), \quad S_{G,\mathrm{cgo}}(t), \quad \text{and} \quad S_{G,\mathrm{cgo}}^\otimes(t). \tag{8}$$

To define the next entropy, we introduce the von Neumann entropy $H_{\mathrm{vN}}(\rho) = -\mathrm{tr}\{\rho \ln \rho\}$ of a density matrix $\rho$. Furthermore, let $\pi_A(t) \sim e^{-\beta_{\mathrm{can}}^A(t) H_A}$ denote the canonical Gibbs state of system $A$ (and similarly for $B$), where the inverse temperature $\beta_{\mathrm{can}}^A(t)$ is chosen such that the coarse energy expectation value of $\pi_A(t)$ matches the one of the true macro-distribution $p_\alpha(t)$, i.e., $\beta_{\mathrm{can}}^A(t)$ is indirectly defined by equating $\sum_\alpha E_\alpha \mathrm{tr}\{\Pi_\alpha^A \pi_A(t)\} = \sum_\alpha E_\alpha p_\alpha(t)$. Then, we define the *canonical entropy*

$$S_{\mathrm{can}}(\langle E \rangle; t) \equiv H_{\mathrm{vN}}[\pi_A(t)] + H_{\mathrm{vN}}[\pi_B(t)]. \tag{9}$$

Recent studies of it include Refs. [67, 76–79].

We remark that the definitions in Eqs. (8) and (9) require knowledge of the DOS *out-side* the dynamically relevant regime, which we have discussed above and colored in red in Fig. 1. Of course, we could simply neglect the red part and only use the green part in Fig. 1 in Eqs. (8) and (9), but this could give very different result (depending on the DOS) and could be unrealistic (as a real body has a non-zero DOS over a wide range of energies).

We continue with the definition of an *entanglement entropy* [80, 81]

$$S_{\text{ent}}(t) \equiv H_{\text{vN}}[\rho_A(t)] + H_{\text{vN}}[\rho_B(t)] = 2H_{\text{vN}}[\rho_A(t)], \tag{10}$$

where $\rho_A(t) = \text{tr}_B\{|\psi(t)\rangle\langle\psi(t)|\}$ is the reduced state of system $A$ (and similarly for $B$). Note that we are interested in the entanglement entropy of the compound system, which is the reason why we add the contribution of $A$ and $B$. Moreover, $H_{\text{vN}}[\rho_A(t)] = H_{\text{vN}}[\rho_B(t)]$ is due to the fact that the state of the compound system is pure.

Finally, we define *diagonal entropy* [34, 35]

$$S_{\text{diag}}(t) \equiv H_{\text{Sh}}[p_j^A(t)] + H_{\text{Sh}}[p_j^B(t)], \tag{11}$$

where $p_j^A = \langle\epsilon_j^A|\rho_A|\epsilon_j^A\rangle$ is the probability to find system $A$ with the microscopic eigenenergy $\epsilon_j^A$ (and similarly for $B$). This concludes the definition of the eleven entropies that we will compare below.

In addition, we will also look at three possible temperature definitions for given expectation values $\langle E_{A,B}\rangle$. First, we introduce the inverse Boltzmann temperature

$$\beta_B(t) \equiv \left.\frac{\partial \ln\mu(\epsilon)}{\partial\epsilon}\right|_{\epsilon=\langle E_A\rangle(t)}. \tag{12}$$

Second, based on $\omega(\epsilon) = \int_{E_{\min}}^{\epsilon}\mu(\epsilon')d\epsilon'$ we introduce the inverse Gibbs temperature

$$\beta_G(t) \equiv \left.\frac{\partial}{\partial\epsilon}\ln\omega(\epsilon)\right|_{\epsilon=\langle E_A\rangle(t)} = \left.\frac{\mu(\epsilon)}{\omega(\epsilon)}\right|_{\epsilon=\langle E_A\rangle(t)}. \tag{13}$$

Note that, while $\beta_B$ can become negative if $\ln\mu(\epsilon)$ is not monotonically increasing, $\beta_G$ is always positive, which is the reason for the controversy in Refs. [29, 69–75].

Third, we consider the temperature we have already implicitly used above to define the canonical entropy in Eq. (9) and consequently call it the canonical inverse temperature $\beta_{\text{can}}(t)$. As explained above, it is obtained from a canonical Gibbs distribution by matching its energy expectation value to the actual value. Note that $\beta_{\text{can}}(t)$ can also be negative. It has been introduced in phenomenological thermodynamics in Refs. [82, 83] and recent studies in statistical mechanics include Refs. [67, 76–79, 84].

Note that all three temperature definitions, together with their respective entropy notions, can be used to establish the Clausius relation $dS = \beta dE$. Moreover, for a broader overview on nonequilibrium temperatures we refer to Refs. [85, 86].

Finally, all entropy notions are summarized in Table 1. Note that they are all based on the common idea to treat the system in a coarse way using incomplete information, just how they do it precisely differs from definition to definition. In contrast, the full microscopic von Neumann entropy $H_{\text{vN}}[\psi(t)]$ is zero and remains constant all the time.

# 3 Numerical results

Below, we use two more conventions beyond what has been discussed above. First, $\alpha, \beta \in \{1, 2, \ldots, 7\}$ labels the seven dynamically relevant energy windows in $[E_a, E_b]$ in increas-

Table 1: Overview of all the different entropy notions compared in this work. We dropped the dependence on time $t$ for notational simplicity.

| name | symbol | definition | comment |
|---|---|---|---|
| Boltzmann entropy of the average | $S_B(\langle E \rangle)$ | $\ln W_{\langle \alpha \rangle} + \ln W_{\langle \beta \rangle}$ | related to $\beta_B$ in the continuum limit $\delta_E \to 0$ |
| averaged Boltzmann entropy | $\langle S_B \rangle$ | $\langle \ln W_\alpha \rangle + \langle \ln W_\beta \rangle$ | |
| cgo entropy | $S_{\text{cgo}}$ | $H_{\text{Sh}}[p_{\alpha,\beta}] + \langle S_B \rangle$ | |
| local cgo entropy | $S_{\text{cgo}}^\otimes$ | $H_{\text{Sh}}[p_\alpha \otimes p_\beta] + \langle S_B \rangle$ | |
| volume entropy of the average | $S_G(\langle E \rangle)$ | $\ln \Omega_{\langle \alpha \rangle} + \ln \Omega_{\langle \beta \rangle}$ | related to $\beta_G$ in the continuum limit $\delta_E \to 0$ |
| averaged volume entropy | $\langle S_G \rangle$ | $\langle \ln \Omega_\alpha \rangle + \langle \ln \Omega_\beta \rangle$ | |
| cgo entropy (volume version) | $S_{G,\text{cgo}}$ | $H_{\text{Sh}}[p_{\alpha,\beta}] + \langle S_G \rangle$ | |
| local cgo entropy (volume version) | $S_{G,\text{cgo}}^\otimes$ | $H_{\text{Sh}}[p_\alpha \otimes p_\beta] + \langle S_G \rangle$ | |
| canonical entropy | $S_{\text{can}}(\langle E \rangle)$ | $H_{\text{vN}}[\pi_A] + H_{\text{vN}}[\pi_B]$ | related to $\beta_{\text{can}}$ |
| entanglement entropy | $S_{\text{ent}}$ | $H_{\text{vN}}[\rho_A] + H_{\text{vN}}[\rho_B]$ | |
| diagonal entropy | $S_{\text{diag}}$ | $H_{\text{Sh}}[p_j^A] + H_{\text{Sh}}[p_j^B]$ | |

ing order (previously, $\alpha, \beta \in \{1, 2, \ldots, M\}$ labeled the energy windows in $[E_{\min}, E_{\max}]$). Second, we plot everything over a dimensionless time $t/\tau$ by introducing a characteristic nonequilibrium time scale $\tau$. To this end, we use Fermi's golden rule $\Gamma_{i \to f} = 2\pi |\langle \epsilon_f | V | \epsilon_i \rangle|^2 \rho(\epsilon_f)$, where $\epsilon_i$ ($\epsilon_f$) denote initial (final) energy eigenstates and $\rho(\epsilon)$ the DOS of the unperturbed Hamiltonian $H_A + H_B$. Owing to the bandedness of the interaction Hamiltonian, the systems needs to make at least $\Delta_E/\delta_V$ many transitions to explore the entire energy range. Thus, we set

$$\tau = \frac{\Delta_E}{\delta_V} \frac{1}{\Gamma_{i \to f}} \approx \frac{\Delta_E^2}{2\pi \lambda^2 \delta_V D_{\text{eff}}^A D_{\text{eff}}^B} . \tag{14}$$

Here, we approximated $|\langle \epsilon_f | V | \epsilon_i \rangle|^2 = \lambda^2$ for $|\epsilon_f - \epsilon_i| \le \delta_V$ (which is true on average) and $\rho(\epsilon_f) \approx D_{\text{eff}}^A D_{\text{eff}}^B / \Delta_E$. The latter approximation is rather crude and completely neglects the fine structure of the DOS. Thus, $\tau$ serves as a convenient but not rigorous estimate of the nonequilibrium time scale.

Moreover, all numerical results are displayed for a single choice of the random matrix Hamiltonian and a single choice for the three types of initial states. We do not perform any averages over them, but we have observed that the coarse behaviour is quite insensitive to the precise choice of Hamiltonian or initial state (not displayed here).

## 3.1 Class A: Two normal systems

We consider two normal systems $A$ and $B$ with identical DOS $\mu(\epsilon) = \exp(6\sqrt{\epsilon})$ as sketched in Fig. 2(A). The effective Hilbert space dimension of the dynamically relevant energy interval $[E_a, E_b]$ is for both systems $D_{\text{eff}}^A = D_{\text{eff}}^B = 397$ (hence, the total Hilbert space dimension is 157,609). The initial macrostate is $p_{\alpha,\beta}(0) = \delta_{\alpha,2}\delta_{\beta,6}$. All numerical results are displayed in Fig. 3.



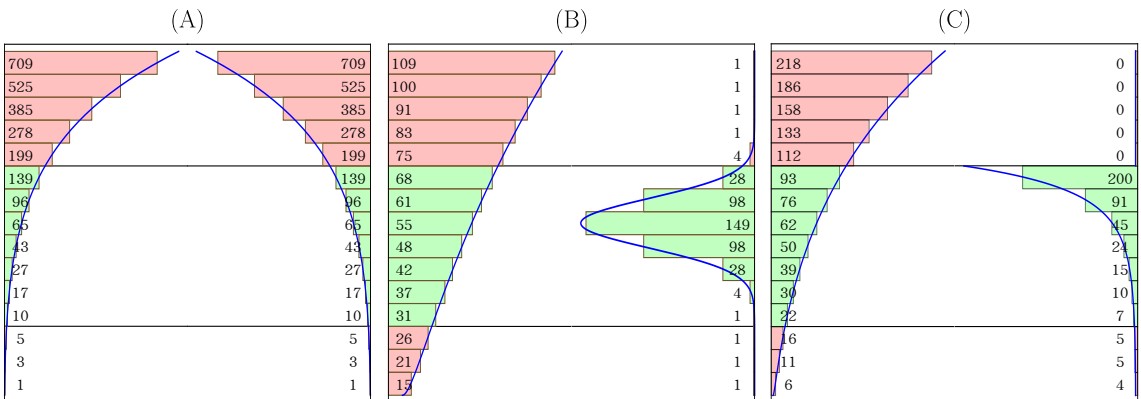

Figure 2: With the same convention as in Fig. 1, we display the smooth DOS $\mu_{A/B}(\epsilon)$ and coarse-grained DOS histogram for systems $A/B$ for the three classes of setups (A), (B) and (C) considered here. Moreover, the numbers in the histogram equal the $W_{\alpha/\beta}$ used in the numerical simulation.

We start by investigating the nonequilibrium time evolution of the local macrostate distributions. To this end, we display a $3 \times 4$ grid of histograms of $p_\alpha(t)$ (dark, blue bars) and $p_\beta(t)$ (bright, orange bars) for different times $t/\tau \in \{0, 10, 20, 30\}$ at the top of Fig. 3 (note that the $y$-axis has the same scale everywhere). Each row of the histogram grid corresponds to one of the three different initial conditions defined above Eq. (2): local energy eigenstates (IC1, first row), decorrelated Haar random states (IC2, second row) and entangled Haar random states (IC3, third row) confined to the microcanonical subspace. The most important thing to notice is that the evolution of the macrostate distribution is almost identical for all three initial conditions (numerical discrepancies in the plot are barely visible to the eye), even though the microstates are *very* different. We here call this property *dynamical typicality* [87–91]. This makes a thermodynamic analysis in terms of macrostates meaningful.

Moreover, we see that the macrostates spread out and do not remain strongly peaked around some mean energy, indicating the need of a probabilistic description. This behaviour is indeed expected for mesoscopic systems for which the need to find a proper definition of entropy and temperature is pressing. It would vanish in a suitable macroscopic limit, which is numerically not accessible to us. Note that any uncertainty in $p_{\alpha,\beta}(t)$ is entirely of quantum origin in our model, there is no classical uncertainty. Finally, we can observe a (mirror) symmetry between $p_\alpha(t)$ and $p_\beta(t)$, as one would expect for two identical systems.

Next, we consider the time evolution of the average energies of $A$ and $B$ in Fig. 3(a) for IC2 and we observe two things. First of all, we see that the quantum expectation value $\langle H_A \rangle(t) = \text{tr}_A\{H_A \rho_A(t)\}$ is all the time very close to the coarse-grained expectation value $\langle E_A \rangle = \sum_\alpha E_\alpha p_\alpha(t)$, and similarly for $B$. This is important because it shows that the widths of the energy windows is chosen well. Second, we see the energies converging to the same value for long times because the two systems are identical.

We continue with the time evolution of the three different inverse temperature definitions in Fig. 3(b) for IC2. We see that $\beta^A(t) \approx \beta^B(t)$ for long times and for all three different definitions, even though the different definitions do not agree among each other. However, it is known that the remaining discrepancy would vanish for larger systems due to the equivalence of ensembles [7–9], but it was numerically not possible to reach this regime.

Finally, we turn to the nonequilibrium dynamics of entropy in Figs. 3(c–e) and remark that the thin horizontal gray line indicates the value $\ln(D_{\text{eff}}^A D_{\text{eff}}^B)$ for orientation. In Fig. 3(c) and Fig. 3(d) we plot the time evolution of the entanglement and diagonal entropy, respectively, for all three different initial conditions. The important point to observe is that both notions

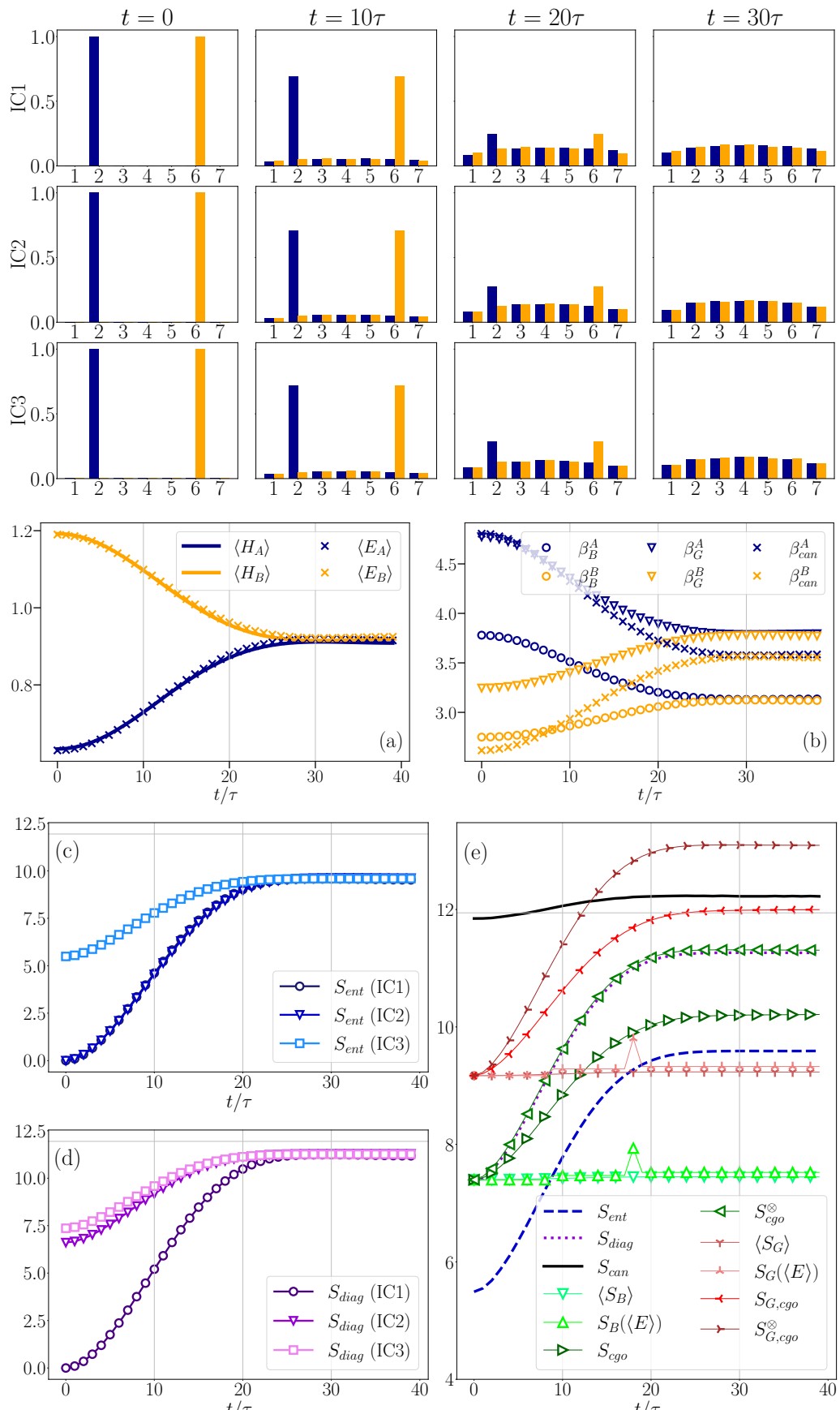

Figure 3: Numerical results for class A (for a description see the main text).

do not depend on the macrostate alone but are sensitive to the microstate for transient times. This implies that also the associated second law, quantified by the change in entanglement or diagonal entropy (the *entropy production*), respectively, would depend on the microstate. Note that this is not the case for any of the other thermodynamic quantities, which is the reason why we plot their evolution for a single initial condition only (here IC3) in Fig. 3(e).

Turning to the comparison of all eleven different entropies in Fig. 3(e), we see that all of them tend to increase, i.e., they satisfy the second law of thermodynamics. However, both the Boltzmann surface and Gibbs volume entropies $S_B(\langle E \rangle)$, $\langle S_B \rangle$, $S_G(\langle E \rangle)$ and $\langle S_G \rangle$ stay almost constant. The upward and downward jumps of $S_B(\langle E \rangle)$ and $S_G(\langle E \rangle)$ are caused by a brief transition of the average energy from one discrete energy window to another. Also the canonical entropy production is significantly smaller than the production of the entanglement, diagonal and cgo entropies. It is further noticeable that the diagonal entropy $S_{\text{diag}}$ (for IC2 and IC3) almost perfectly matches the local cgo entropy $S_{\text{cgo}}^{\otimes}$, while the entanglement entropy $S_{\text{ent}}$ (for IC3) evolves in parallel but not very close to the cgo entropy $S_{\text{cgo}}$. Moreover, as it should be for normal systems, all volume-related entropies show the same behaviour, just shifted upwards to larger numerical values (that is, the entropy productions are the same if we interchange the letters $B$ by $G$).

Finally, it is worth to notice that von Neumann's $H$-theorem [4,5] (see also Refs. [92–95]) establishes the conditions under which the cgo entropy $S_{\text{cgo}}$ saturates to its maximum value $\ln(D_{\text{eff}}^A D_{\text{eff}}^B)$ (thin horizontal gray line), which does not happen in our example. We attribute this discrepancy to the fact that the perturbation $V = \lambda R$ is too banded or, equivalently, the global state is not confined to a sufficiently narrow energy shell. In this case, the eigenfunctions of $H_A + H_B + V$ are not delocalized over the entire dynamically relevant Hilbert space (we can not prove this fact as an exact diagonalization of the total Hamiltonian is out of reach) such that the resulting non-negligible correlations between the global and local energy eigenbasis could jeopardize the applicability of von Neumann's $H$-theorem.

## 3.2 Class B: Normal system coupled to a negative temperature system

We continue by considering a normal system $A$ with DOS $\mu_A(\epsilon) = \exp(\sqrt{3\epsilon})$ coupled to a system with a Gaussian DOS $\mu_B(\epsilon) = \exp[-(\epsilon - \bar{\epsilon})^2/\sigma^2 + c]$, where we set the mean and variance to $\bar{\epsilon} = (E_b + E_a + 2\delta_E)/2$ and $\sigma = (E_b + E_a)/12$, respectively, and introduced a constant $c = (E_{\text{max}} - \bar{\epsilon})^2/(2\sigma^2)$. For a sketch see Fig. 2(B). A Gaussian DOS would naturally arise (at least approximately) for spin systems and it is characterized by the appearance of negative Boltzmann temperatures in the regime where $\partial_\epsilon \ln \mu_B(\epsilon) < 0$. The correct thermodynamic treatment of such systems has caused some debate [29, 69–75]. The initial macrostate is $p_{\alpha,\beta}(0) = \delta_{\alpha,2}\delta_{\beta,6}$ and the effective dimensions are $D_{\text{eff}}^A = 342$ and $D_{\text{eff}}^B = 406$. All numerical results are displayed in Fig. 4 using the same conventions as for Fig. 3.

We start again by discussing the nonequilibrium dynamics of the local macrostate distributions ($3 \times 4$ grid of histograms at the top of Fig. 4). As before, we see dynamical typicality. Moreover, system $B$ has a clearly visible preference for occupying the energy window $E_{\beta=5}$ with the largest amount of states, as expected.

The evolution of the local average energies is shown in Fig. 4(a), demonstrating again that the coarse-grained average energy matches well the exact quantum expectation value. Now, however, the average energy of $A$ and $B$ do not tend to the same value (which is also visible in the histograms) since the systems are not identical.

The inverse Boltzmann and canonical temperature tend to converge to a similar value for long times as shown in Fig. 4(b). The remaining small but noticeable difference for long times is attributed to finite size effects. Instead, the difference in Gibbs temperature seems not attributable to finite size effects. In fact, one even observes that the difference in Gibbs temperature *increases* in time (the coincidence at $t = 0$ is accidental).

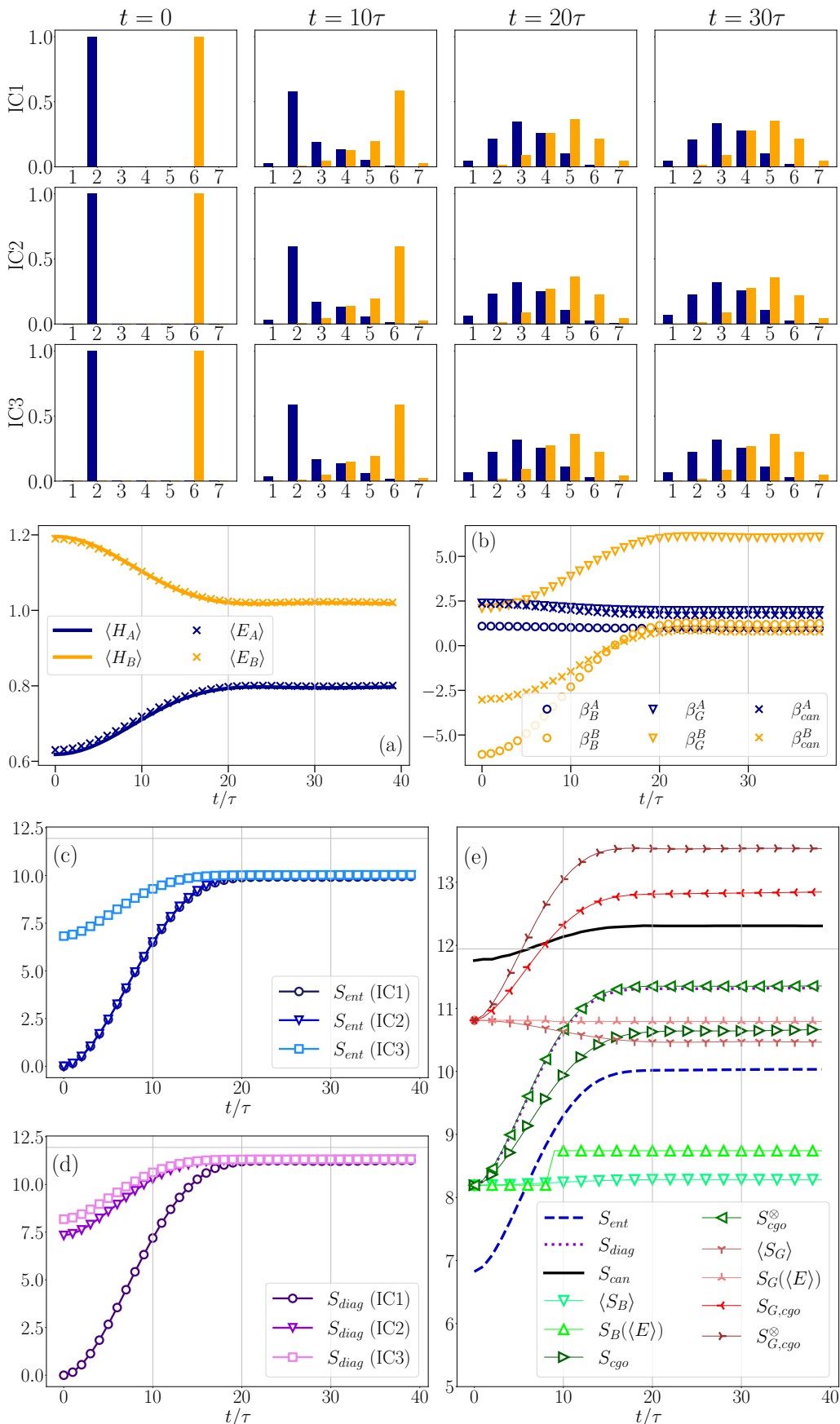

Figure 4: Numerical results for class B (for a description see the main text).

Finally, we turn to the nonequilibrium dynamics of entropy in Fig. 4(c–e). The qualitative features are identical to class A with the only important difference that the Gibbs-volume entropies $S_G(\langle E\rangle)$ and $\langle S_G\rangle$ now violate the second law.

### 3.3   Class C: Normal system coupled to a negative heat capacity system

Finally, we consider a normal system $A$ with DOS $\mu_A(\epsilon) = \exp(\sqrt{10\epsilon})$ coupled to a system $B$ with DOS $\mu_B(\epsilon) = \Theta(E_b - \epsilon)\exp[(3\epsilon/2)^2]$. Since $C \sim -\partial_\epsilon^2 \ln \mu_B(\epsilon) = -3/2$ this describes a system with negative heat capacity as it could arise in systems with long range interactions [10–15]. Moreover, the Heaviside step function $\Theta(x)$ introduces a cutoff outside the dynamically relevant regime (physically, this could be justified by a gaped DOS or an additional conservation law). The resulting situation is sketched in Fig. 2(C). The initial macrostate is $p_{\alpha,\beta}(0) = \delta_{\alpha,1}\delta_{\beta,7}$ and the effective dimensions are $D_{\mathrm{eff}}^A = 372$ and $D_{\mathrm{eff}}^B = 392$. All numerical results are displayed in Fig. 5 with the same conventions as in Figs. 3 and 4.

As before, the grid of histograms at the top of Fig. 5 displays the time evolution of the macrostates, affirming dynamical typicality. Furthermore, there is a clear tendency to reside in the state with the highest Boltzmann entropy despite some smearing out.

Figure 5(a) displays the evolution of the average energies, giving rise to the same observation as for class B.

Figure 5(b) shows that both the Boltzmann and Gibbs temperature face problems. One could say that the Gibbs temperature performs a bit better, but in general both temperatures of $A$ and $B$ decrease, describing a spontaneous cooling of the entire setup. However, such strange behaviour is expected for systems with negative heat capacity and does not necessarily violate the second law (but see below). Remarkably, the canonical temperature shows a clear tendency to approach each other, even though it does not reach the same value. Moreover, observe that for the present initial state and DOS the initial inverse temperature $\beta_{\mathrm{can}}^B(0)$ of system $B$ is very negative.

Finally, we conclude with a study of the entropies in Fig. 5(c–e). We can now see that both the Boltzmann-surface entropies $S_B(\langle E\rangle)$ and $\langle S_B\rangle$ and the Gibbs-volume entropies $S_G(\langle E\rangle)$ and $\langle S_G\rangle$ clearly violate the second law (the Boltzmann entropy even stronger than the Gibbs entropy). Moreover, in contrast to class B and C, the canonical entropy increases significantly, which is related to the just mentioned strong negativity of $\beta_{\mathrm{can}}^B(0)$. Note that the behaviour of canonical entropy depends sensitively on the behaviour of the DOS outside the dynamically relevant regime: if instead of the Heaviside step function one allows some small $\mu_B(\epsilon) > 0$ for $\epsilon > E_b$ one quickly restores a behaviour similar to class A and B (not shown here). Otherwise, we observe a similar behaviour for all the other entropies: they are clearly increasing, $S_{\mathrm{ent}}$ and $S_{\mathrm{diag}}$ depend on the microstate, and $S_{\mathrm{diag}}$ (for IC3) and $S_{\mathrm{cgo}}^\otimes$ almost perfectly coincide.

## 4   Summary

Before we summarize our observations, we briefly discuss the weaknesses and strengths of the model chosen here.

The weaknesses are related to the particularity of the model: a random matrix Hamiltonian (in reality no Hamiltonian is truly random), the arbitrary choice of DOS for $H_A$ and $H_B$ (which we have not obtained from any explicit model), the choice of initial states that are strongly localized in energy (which would not result from a prior coupling of $A$ or $B$ to heat reservoirs), as well as other compromises forced on us by numerical limitations. However, the present model has also several strengths that tame the criticism.

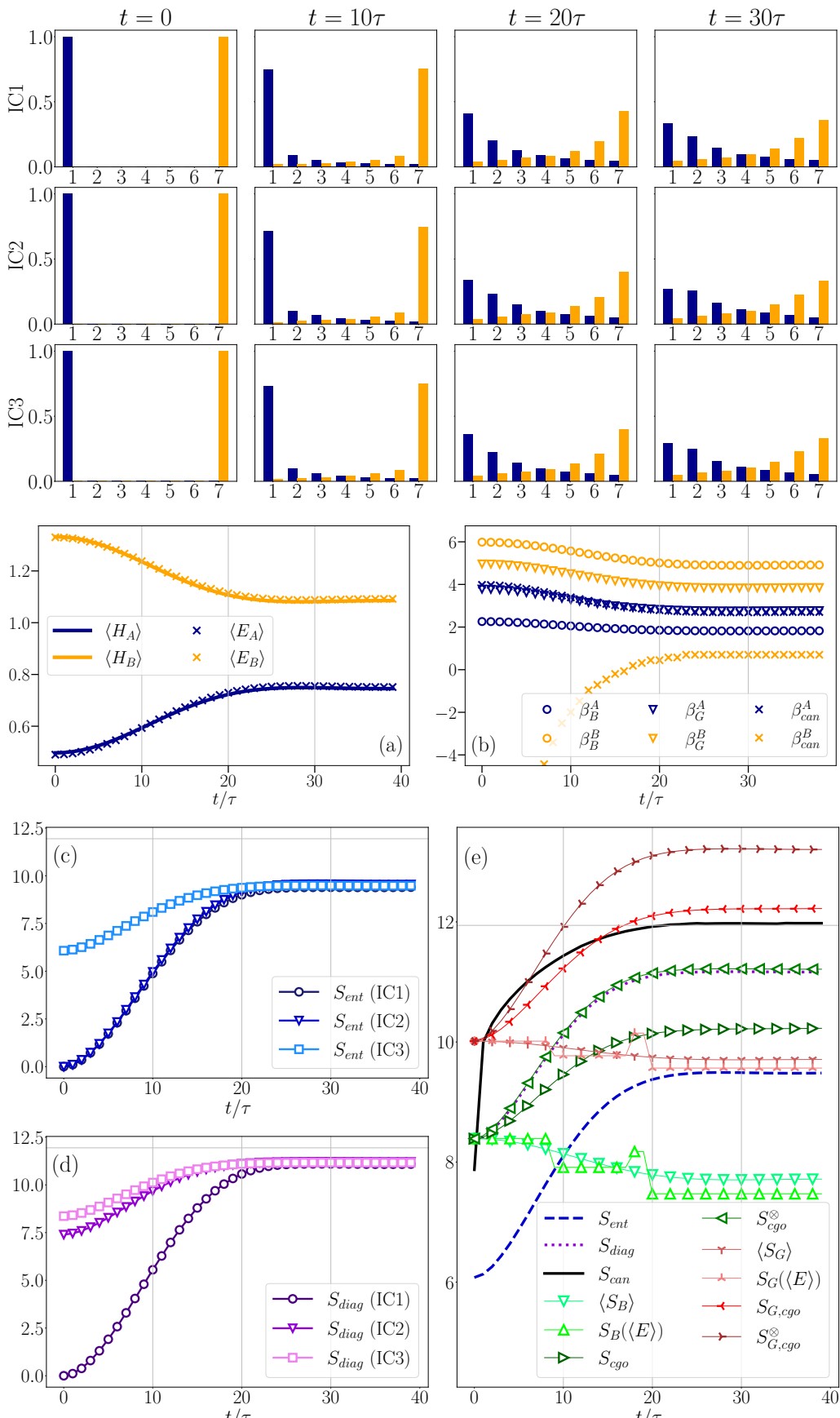

Figure 5: Numerical results for class C (for a description see the main text).

First, while we presented numerical results only for a single choice of the random matrix interaction $V = \lambda R$ and a single random choice of the three different initial conditions, we have tested the code extensively and observed quantitatively very similar behaviour in each run. Thus, the success of random matrix theory [54–60] and the closely related eigenstate thermalization hypothesis [59–64] makes us confident that our model captures at least qualitatively some relevant features of realistic heat exchange models. Similarly, we have also checked that many qualitative features remain when changing the local DOS slightly.

Second, while our choice for the initial state is not inspired by a prior coupling of $A$ or $B$ to separate heat reservoirs, one can imagine other experimental procedures that prepare $A$ or $B$ within a microcanonical energy window. For instance, one could initially measure the coarse energy of $A$ or $B$, or one could start from the ground state and then inject fixed amounts of energy. Moreover, it seems questionable that a more smeared out initial distribution would result in great changes of our results. At least the entropies $S_B(\langle E \rangle)$, $\langle S_B \rangle$, $S_G(\langle E \rangle)$, $\langle S_G \rangle$ and $S_{\text{can}}$ are insensitive to the variance. We found microcanonical initial conditions more interesting as canonical initial states are conventionally considered in many fields (e.g., in Refs. [23–25]).

It is also worth mentioning that scaling up the local Hilbert space dimensions but leaving the DOS and coarse-graining fixed would not result in qualitatively different results because the qualitative features of the dynamics depend on the relative ratio of the volumes $W_\alpha$ and $W_\beta$ but not their absolute values.

Finally, if the second law and the entropy concept truly are universal, as it is widely proclaimed, they should certainly uphold a test within a simple toy model that captures some elementary aspects of a heat exchange model.

We now summarize our findings.

1. Both entanglement and diagonal entropy sensitively depend on the initial microstate even though the macrostates show the same dynamics. If one chooses the third initial condition, which corresponds to the maximally unbiased pure state with respect to the initial macrostate, the evolution of diagonal entropy becomes virtually identical to the evolution of local cgo entropy and the evolution of entanglement entropy shows some similarity with the evolution of cgo entropy.

2. Boltzmann's surface entropy concept, whether we evaluate it for the average energy or consider its average, strongly violates the second law for class C (a normal system coupled to a negative heat capacity system). Also the associated temperature notions shows no tendency of convergence then.

3. Gibbs' volume entropy concept, whether we evaluate it for the average energy or consider its average, violates the second law for class B (a normal system coupled to a negative temperature system) *and* C. Also the associated temperature notions shows no tendency of convergence then.

4. For classes A and B, the canonical entropy increases only very little in comparison to the entanglement, diagonal and cgo entropies, but for class C it increased strongly. This behaviour is related to the sensitivity of canonical entropy with respect to the DOS outside the dynamically relevant regime. Its associated temperature notion performs very well for class A and B and shows some tendency of convergence for class C.

5. Cgo entropies, whether of surface or volume type, show a strong increase similar to the entanglement or diagonal entropy. This highlights that probabilistic considerations, captured by a Shannon entropy term in cgo entropies, seem important for the validity of the second law. In particular, we repeat that these probabilities are genuinely of quantum origin in our model: they do not arise from classical ensemble averages or a subjective

lack of classical knowledge. Finally, despite its increase, von Neumann's $H$-theorem for cgo entropy $S_{cgo}$ does not apply, presumably because of the too narrow band structure of $V$ (or the too widespread energy interval) considered here.

In spirit of our paper, we leave it to the reader to draw further conclusions from these numerical observations.

# Acknowledgments

**Funding information** Financial support by MICINN with funding from European Union NextGenerationEU (PRTR-C17.I1) and by the Generalitat de Catalunya (project 2017-SGR-1127) are acknowledged. PS is further supported by "la Caixa" Foundation (ID 100010434, fellowship code LCF/BQ/PR21/11840014), the European Commission QuantERA grant Ex-TRaQT (Spanish MICIN project PCI2022-132965), and the Spanish MINECO (project PID2019-107609GB-I00) with the support of FEDER funds.

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
