# Peer review of "Comparative Microscopic Study of Entropies and their Production"

_SciPost Physics, doi:SciPost Phys. 17, 143 (2024)_

## Round 1 · Referee Report · Francesco Buscemi (Referee 1) · 2024-4-28

Strengths

1- The paper is very well written. 2- The strengths and weaknesses of the numerical analysis presented are honestly explained.

Weaknesses

1- I didn't find any particular weaknesses, other than those listed by the authors themselves, and I don't think they affect the results presented.

Report

I enjoyed reading this work: it's solid science and a useful, if not exactly thrilling, addition to the debate about micro and macro entropies. The message it wants to convey is clearly explained and ultimately delivered in a clear way, as expected. I am happy to recommend it for publication. (But see requested changes below.)

Requested changes

1- The story that von Neumann suggested the name "entropy" to Shannon with the infamous tongue-in-cheek comment seems to be a factoid rather than a fact; Shannon denied it ever happened. See https://www.eoht.info/page/Neumann-Shannon%20anecdote 2- I would find it very useful to have all the entropies and cases considered summarized in some sort of table. 3- I would also find it useful if the Authors made their code available for other people to inspect and use. I believe this is also required by this journal's policies (see point number 6 in the "General required acceptance criteria")

Recommendation

Ask for minor revision

  • validity: top
  • significance: good
  • originality: good
  • clarity: high
  • formatting: excellent
  • grammar: perfect

Author:  Philipp Strasberg  on 2024-05-31  [id 4527]

(in reply to Report 1 by Francesco Buscemi on 2024-04-28)

Thank you for the constructive and positive report. We made various small improvements and followed your suggestions. Specifically:

  1. That's an interesting point and we are now more careful here. However, to be fair, in the interview it seems also that Shannon's memory might not be the best anymore. So perhaps the question about the truth of this quote will never be settled.

  2. That's a good point and we now included such a table at the end of Sec. 2.

  3. That's another fair point. While submitting the paper, we didn't see any option for submitting the code, but we might have overlooked it. Our code is a combination of Mathematica (used to generate some parameters/functions and evaluate the results) and Matlab (used to build the sparse Hamiltonian and time evolve the state) and we use an ordinary desktop computer with 32 GB of memory to run it. We did not think about sharing the code because it didn't appear very sophisticated to us, but we are happy to share it upon request or include it in the submission if that is the preference of the editor (and if they tell us how to do it).

---

## Round 1 · Referee Report · Valerio Scarani (Referee 2) · 2024-5-10

Strengths

  1. The debate on the various definitions of entropy is tackled by studying their behavior in an actual, reasonable and simple model; rather than by discussing their formal properties. This is a novel approach and a very valid addition to the literature.
  2. The writing is clear, the assumptions and positions of the authors are clearly stated.

Weaknesses

None that I can obviously see.

Report

This paper deals with the proliferation of definitions of thermodynamical entropy that can be found in the literature. The authors explicitly choose not to participate in the ongoing debate by commenting on properties of those definitions, but rather let their predictive power speak, by checking how each definition fares on one of the most elementary thermodynamical situations: the exchange of energy between two systems.
Initially, the model consists of two systems with Hamiltonian interaction, without dissipative terms. The apparent thermodynamical behavior may come from a variety of added features: (a) the coarse-graining of the knowledge; (b) the restriction to some energy levels (while others may be coupled, thus leading to “dissipation” into those); (c) the choice of a “random matrix” for the coupling; (d) the entanglement generated, compounded with a definition of entropy that uses only local information; (e) of course, interplay among some of these features.
The behaviour of the definitions on this model is then tested on various types of systems, and various types of initial states; the observed features are described, with an emphasis on possible violations of the second law.
The authors make a great effort to present their assumptions and motivate them. To comment on the results, we would have had to redo the numerical simulations, which we did not: in that respect, we trust that the reported conclusions are correct.
Overall, we think that this paper is important, as it opens a new way of looking at the debate. Surely it will trigger several follow-ups and discussions. It should be accepted after considering the revisions suggested below.

Requested changes

  1. The most technical point first: it is not clear to us if the interaction V used in each simulation is fixed (one “random matrix” has been chosen); or if it is itself chosen at random in every simulation, and the reported results are an average over those choices.
  2. Around Eq. (1): why does it matter that “the sum of the local energies is approximately conserved”?
  3. Eq. (1) comes out of the blue, besides the fact that it was “found to work well”. Can this finding process be clarified? Incidentally, there is a typo on the lhs: \delta V instead of \delta_V.
  4. In commenting Fig. 3, the “short upward and downward fluctuation” is said to be “related to the discrete coarse graining”. Maybe this is obvious for those who have gone through coding these things, but for us it is not. If it could be clarified without great effort, it would be nice (if not, no problem, it is a minor point and we are ready to accept some numerical glitch as the origin of that feature).
  5. Most failures to satisfy the Second Law are reported for a normal system coupled to a system with negative heat capacity. First of all, it would be good to list a few examples of such systems (as was done for the negative temperature case). More about the results: our intuition on such systems is weak: for instance, if both systems start at the same temperature, the evolution may be that both temperatures increase (the negative heat capacity system losing heat to the positive heat capacity one). In that case, is it possible that some of the entropies that “violate the second law” are actually capturing some of these counter-intuitive features? Then, rather than being problematic, they would be actually sensing the system better than the others.
  6. If possible, it would be good to clarify more specifically which of the features (a)-(e) listed above are relevant as “paths to thermodynamics”. We presume that some may matter more than other, depending on the entropy. Or said differently: maybe some entropy definitions are not suited to capture a given type of loss of information, leading to the paradoxes.
  7. Last: this paper is clearly written to stimulate the debate among experts. Sure enough, we don't expect a review paper; and references are given for those who want to study. Still, having something like a table where the different definitions of entropy are listed, alongside with their most famous properties, or use cases, would facilitate the reading for the non-initiated.

Recommendation

Ask for minor revision

  • validity: high
  • significance: top
  • originality: high
  • clarity: high
  • formatting: excellent
  • grammar: excellent

Author:  Philipp Strasberg  on 2024-05-31  [id 4526]

(in reply to Report 2 by Valerio Scarani on 2024-05-10)

We thank you for the positive and constructive report. We have made small changes throughout the manuscript and mostly followed your suggestions. To comment specifically on your requested changes:

  1. Thanks for spotting that we haven't been clear here. All plots present results for a single random matrix (and a single realization for the three types of random initial states). We have emphasized this now in the paragraph before Sec. 3.1.

  2. This is done for physical reasons: by avoiding that the interaction can store too much energy, we remain in the relevant weak coupling regime where locally defined temperatures, energies, entropies for system A or B remain meaningful.

  3. It is true that we have been a bit quick here. Getting the parameters right in a finite size system is a non-trivial problem that involves some compromises. We have now been more explicit at this point to better explain our reasoning.

  4. This is due to the discrete nature of the coarse graining, making the Boltzmann/Gibbs entropy of the average evolving discontinuously compared to the other entropies. We reformulated the sentence at this point.

  5. Negative heat capacities often arise in systems with long range interactions (e.g., a black hole) and it is quite true that these systems behave in a bizarre way. It is therefore an interesting comment whether entropies violating the second law would be more informative here, but this discussion appears to us too speculative for the present paper. We think the right take away message is here: entropies (if you choose the right one) remain meaningful, but temperatures (most likely) do not. Thus, Clausius' formulation that heat flows from hot to cold is simply wrong in general even for macroscopic systems.

  6. We included a small comment in the last paragraph of Sec. 2 on this, but we want to avoid any longer discussion about the (origin of the) qualitative differences between the different definitions. We believe this is in spirit of our paper where we try to focus only on exact numerical results. Thus, we rather see the paper as a stimulating source to think further about the question why the entropies behave as they do, as you already did.

  7. We think it is a good idea to include such a table, which we now did at the end of Sec. 2. However, for space reasons and in spirit of our paper, we only summarize the definitions there and do not list any properties of the entropies.

---

## Round 2 · Referee Report · Francesco Buscemi (Referee 1) · 2024-10-7

Report

I thank the authors for taking my comments into account in this revised version and I am satisfied with the changes.

Recommendation

Publish (meets expectations and criteria for this Journal)

---

## Round 2 · Referee Report · Valerio Scarani (Referee 2) · 2024-10-9

Report

I am happy with the modifications made by the authors.

Recommendation

Publish (easily meets expectations and criteria for this Journal; among top 50%)

---

## Round 2 · List of Changes

Please see the replies to the Referee reports.

---

## Editorial Decision

published